**Data Availability Statement:** This study used national databases obtained from the Health and Welfare Data Science Center (HWDC), Ministry of Health and Welfare in Taiwan (https://dep.mohw.

# Re-examination of the risk of autoimmune diseases after dengue virus infection: A population-based cohort study

Hsin-I Shih[ID][1,2,3], Chia-Yu Chi[4,5], Pei-Fang Tsai[3,6], Yu-Ping Wang[3,4], Yu-Wen Chien[3,7]*

1 Department of Emergency Medicine, National Cheng Kung University Hospital, College of Medicine, National Cheng Kung University, Tainan, Taiwan, 2 School of Medicine, College of Medicine, National Cheng Kung University, Tainan, Taiwan, 3 Department of Public Health, College of Medicine, National Cheng Kung University, Tainan, Taiwan, 4 National Mosquito-Borne Diseases Control Research Center, National Health Research Institutes, Zhunan, Taiwan, 5 Department of Microbiology & Immunology, College of Medicine, National Cheng Kung University, Tainan, Taiwan, 6 Department of Pathology, National Cheng Kung University Hospital, College of Medicine, National Cheng Kung University, Tainan, Taiwan, 7 Department of Occupational and Environmental Medicine, National Cheng Kung University Hospital, College of Medicine, National Cheng Kung University, Tainan, Taiwan

* yuwenchien@mail.ncku.edu

## Abstract

Previous studies suggested that dengue was associated with an increased risk of several autoimmune diseases. However, this association still needs to be explored due to the limitations of these studies. A population-based cohort study was conducted using national health databases in Taiwan and included 63,814 newly diagnosed, laboratory-confirmed dengue patients between 2002 and 2015 and 1:4 controls (n = 255,256) matched by age, sex, area of residence and symptom onset time. Multivariate Cox proportional hazard regression models were used to investigate the risk of autoimmune diseases after dengue infection. Dengue patients had a slightly higher risk of overall autoimmune diseases than non-dengue controls (aHR 1.16; P = 0.0002). Stratified analyses by specific autoimmune diseases showed that only autoimmune encephalomyelitis remained statistically significant after Bonferroni correction for multiple testing (aHR 2.72; P < 0.0001). Sixteen (0.025%) dengue patients and no (0%) controls developed autoimmune encephalomyelitis in the first month of follow-up (HR >9999, P < 0.0001), but the risk between groups was not significantly different thereafter. Contrary to previous studies, our findings showed that dengue was associated with an increased short-term risk of a rare complication, autoimmune encephalomyelitis, but not associated with other autoimmune diseases.

## Author summary

Dengue fever is a mosquito-borne tropical disease caused by the dengue virus. Antibodies produced during dengue infection can cross-react with several self-antigens to the tissues of the body and contribute to thrombocytopenia, coagulopathy, and vascular leakage in severe dengue. Autoimmunity might occur after dengue infection and cause some

gov.tw/dos/cp-5119-59201-113.html). All data obtained were anonymized and deidentified by the HWDC. The data used in this study must be accessed and analyzed in the HWDC after filling out an application according to the relevant regulations and thus cannot be shared. Contact information for data application, analysis and inquiry (https://dep. mohw.gov.tw/dos/cp-2516-59203-113.html).

**Funding:** This study was partially supported by grants from the Ministry of Science and Technology, Taiwan (MOST 107-2314-B-006 -075 -MY3[YWC], MOST 110-2625-M-006 -009- [HIS], MOST 111-2625-M-006-016 [HIS]), National Health Research Institute (MR-108-GP-03 [CYC], MR-110-GP-03 [CYC], and MR-111-GP-05 [CYC]), and National Cheng Kung University Hospital (NCKUH-11103007[HIS]. The funders had no role in study design, data collection and analysis, decision to publish, or preparation of the manuscript. Funder's website: Ministry of Science and Technology, Taiwan: https://www.most.gov. tw/ National Health Research Institutes: https:// www.nhri.edu.tw/.

**Competing interests:** The authors have declared that no competing interests exist.

autoimmune diseases. Previous studies suggested some autoimmune diseases might be associated with dengue infection. Contrary to previous studies, our findings showed that dengue was associated with an increased short-term risk of a rare complication, autoimmune encephalomyelitis, but not associated with other autoimmune diseases.

## Introduction

Dengue is an important mosquito-borne viral disease that has seen a dramatic increase in global incidence in recent decades, probably due to global warming and climate change, rapid urbanization, and the growing frequency of travel and trade [1]. Approximately four billion people are at risk of dengue virus (DENV) infection, particularly in tropical and subtropical countries [2]. It has been estimated that 390 million people are infected by DENV annually, of which 96 million are symptomatic with varying severity [3].

Previous studies also demonstrate that antibodies produced during DENV infection can cross-react with several self-antigens in humans, such as plasminogen, integrin, and platelet cells [4,5]. These autoantibodies can result in autoimmunity and possibly contribute to a clinical progression to severe dengue [4,6]. Although this autoimmunity has been considered transient, a small follow-up study in Cuba showed that over half of dengue patients suffered from persistent dengue symptoms in the two years following infections, and commonly showed altered autoimmune markers, such as antinuclear antibody [7]. Some case reports also suggest that dengue may be associated with several autoimmune diseases [8–12].

Recently, Li *et al.* conducted a population-based cohort study using the National Health Insurance Research Databases (NHIRD) in Taiwan, identifying 12,506 patients who were hospitalized with a diagnosis of dengue fever between 2000 and 2010 and selecting frequency-matched controls for comparison [13]. Their results revealed that dengue patients had a significantly higher risk of developing more than twenty autoimmune diseases within three years after infection (adjusted hazard ratio 1.88; 95% CI, 1.49–2.37), especially post-infectious arthritis, multiple sclerosis, autoimmune encephalomyelitis, systemic vasculitis, systemic lupus erythematosus (SLE), and primary adrenocortical insufficiency [13]. Although that study provided epidemiological evidence for the association between dengue and autoimmune diseases, one of its major limitations was that hospitalized dengue patients with a diagnosis of dengue might not be laboratory-confirmed. Since clinical symptoms of dengue are non-specific, laboratory tests are required to verify the diagnosis, which might take days or even weeks because rapid diagnostic tests were not available in Taiwan during their study period. Recently, the Taiwan Centre for Disease Control (Taiwan CDC) released the Notifiable Disease Dataset of Confirmed Cases (NDDCC) to be linked to the NHIRD. We obtained the list of laboratory-confirmed dengue patients from the dataset and found that only 51.4% of hospitalized patients with a discharge diagnosis of dengue between 2000 and 2010 were finally laboratory-confirmed to have DENV infection. This misclassification bias makes the results from Li *et al.*'s study uncertain [13]. In addition, most dengue patients only have mild symptoms and are treated at outpatient clinics; thus, the inclusion of only hospitalized dengue patients in their study made their results not generalizable to most dengue patients.

Furthermore, southern Taiwan experienced the most severe dengue epidemics in its history in 2014 and 2015, causing over 58,000 confirmed dengue cases. Since autoimmune diseases are relatively rare, including more dengue cases for follow-up will help better elucidate the association between dengue and autoimmune diseases. The objective of this study was to use

laboratory-confirmed dengue cases and to include patients diagnosed before the end of 2015 in Taiwan to re-examine the risk of autoimmune diseases after DENV infection.

## Methods

### Ethics statement

The national databases used in this study were obtained and analyzed at the Health and Welfare Data Science Center (HWDC), Ministry of Health and Welfare of Taiwan. This study was approved by the Institutional Review Board of the National Cheng Kung University Hospital (approval no. B-ER-106-184). All the data were deidentified, and thus, the requirement for informed consent was waived.

### Data sources and study populations

This retrospective cohort study utilized the NHIRD in Taiwan, which included detailed medical insurance claims data from the National Health Insurance (NHI) program since 1995, covering >99% of the population in Taiwan [14]. Through encrypted individual identification numbers, the personal information in the NHIRD could be linked to multiple national databases. The high coverage rate and comprehensiveness of the NHIRD in Taiwan provided valuable resources for long-term follow-up epidemiological studies [14].

In Taiwan, by law, suspected dengue patients should be reported to the local health department and Taiwan CDC within 24 hours of clinical diagnosis. Blood samples from suspected cases should be tested by approved laboratories to confirm the diagnosis. The laboratory criteria of DENV infection included virus isolation, positive real-time reverse transcription-polymerase chain reaction (RT–PCR), detection of dengue-specific IgM and IgG antibody in a single serum sample in the acute phase, four-fold increase in IgG titer in paired acute- and convalescent-phase samples, or detection of the non-structural protein 1 (NS1) in the 2015 dengue epidemic [15,16].

In this study, newly diagnosed dengue patients with laboratory confirmation between 2002 and 2015 were identified from the NDDCC. By defining the index date of a dengue case as the date of symptom onset, four non-dengue controls were randomly selected for each dengue case by matching age, sex, area of residence (Tainan, Kaohsiung, Pingtung, and others), and the calendar year and month of the index date. In the dengue and non-dengue groups, people who were diagnosed with autoimmune diseases before the index date or had missing basic information were excluded. As done by Li *et al.* to minimize confounding by other infections [13], people with a diagnosis of HIV or TB before the index date or having bacterial, viral, or other infections within 60 days before the index date were also excluded. These diseases were defined as at least three outpatient visits or one hospital admission with relevant ICD-9-CM codes, as shown in S1 Table.

### Study outcome and follow-up

The outcome of interest in this study was newly diagnosed autoimmune diseases after the index dates. In Taiwan, people with several autoimmune diseases can apply for catastrophic illness certification to receive a copayment exemption from the NHI program, and all the applications are reviewed by rheumatologists. These autoimmune diseases, including SLE, systemic sclerosis, Sjögren's syndrome, inflammatory myopathy, rheumatoid arthritis, Behcet's syndrome, systemic vasculitis, type I diabetes mellitus, multiple sclerosis, myasthenia gravis, inflammatory bowel diseases, autoimmune hemolytic anemia, and pemphigus, were identified from the Registry for Catastrophic Illness, and thus the diagnoses should be valid. For

autoimmune diseases that did not fulfill the requirements to be considered catastrophic ill-nesses in Taiwan, including ankylosing spondylitis, post-infectious arthritis, uveitis, psoriasis, autoimmune thyroid disease, primary adrenocortical insufficiency, Guillain–Barré syndrome, autoimmune encephalomyelitis, and celiac disease, one hospital admission or at least three outpatient visits with relevant ICD-9-CM or ICD-10-CM codes were regarded as the con-firmed diagnosis (S1 Table). The follow-up of study participants started on the index dates and continued until the confirmation of study outcomes, death, or August 31, 2018, whichever occurred first.

## Covariates

The baseline characteristics considered in this study included age, sex, monthly income, area of residence, and comorbidities. Dengue outbreaks mainly occurred in southern Taiwan, including Tainan, Kaohsiung, and Pingtung. The urbanization level of townships was also adopted as a sociodemographic variable, in which one was the most urbanized and four was the least urbanized, as previously described [15,17]. Comorbidities considered as potential confounders in this study included cerebrovascular accident, chronic obstructive pulmonary disease (COPD), diabetes mellitus (DM), dyslipidemia, hypertension, ischaemic heart disease, liver cirrhosis, malignancy, and renal failure; these coexisting conditions were defined as long as the corresponding ICD-9-CM codes were recorded in at least three outpatient visits or any hospital admission before the index date (S1 Table).

## Statistical analysis

We assessed the differences in baseline characteristics between the dengue and non-dengue groups using the standardized mean differences (SMD), for which a number under 0.1 sug-gests no meaningful difference [18]. The cases of overall and specific autoimmune diseases during the follow-up period were counted separately and divided by person-years to obtain the incidence rates for different outcomes after DENV infection. Kaplan–Meier curves were plotted for the cumulative incidence of autoimmune disease in the dengue and non-dengue groups, and the log-rank test was used for comparison. The association between DENV infec-tion and the occurrence of autoimmune diseases was estimated by hazard ratios (HRs) and 95% confidence intervals (CIs) that were obtained from the Cox proportional hazard model to account for the competing risk of death adjusted for gender, age, sex, residential area, level of monthly income, degree of urbanization, and comorbidities. Subgroup analyses stratified by sex were also performed. The proportional hazard assumption was checked by using log-log survival plots. To account for multiple comparisons for many outcomes and stratified analyses by sex, a Bonferroni post-hoc correction was used to derive an adjusted threshold for p values. Data analysis was conducted using SAS 9.4 (SAS Institute, Cary, NC).

## Results

A total of 319,070 subjects were included in this study, including 63,814 laboratory-confirmed dengue patients and 255,256 matched non-dengue controls (Fig 1). The mean follow-up period was 4.57 ± 3.45 years for the dengue group and 4.56 ± 3.42 years for the non-dengue group. The baseline characteristics for dengue and non-dengue cohorts showed no meaningful difference except that a higher percentage of dengue patients lived in urban areas (Table 1). The distribution of comorbidities was not significantly different between the two groups.

The overall incidence rate of autoimmune diseases in the dengue group was 27.73 per 10,000 person-years, which was slightly higher than the 23.47 per 10,000 person-years in the non-dengue group (Table 2). Dengue patients also had a higher incidence of autoimmune

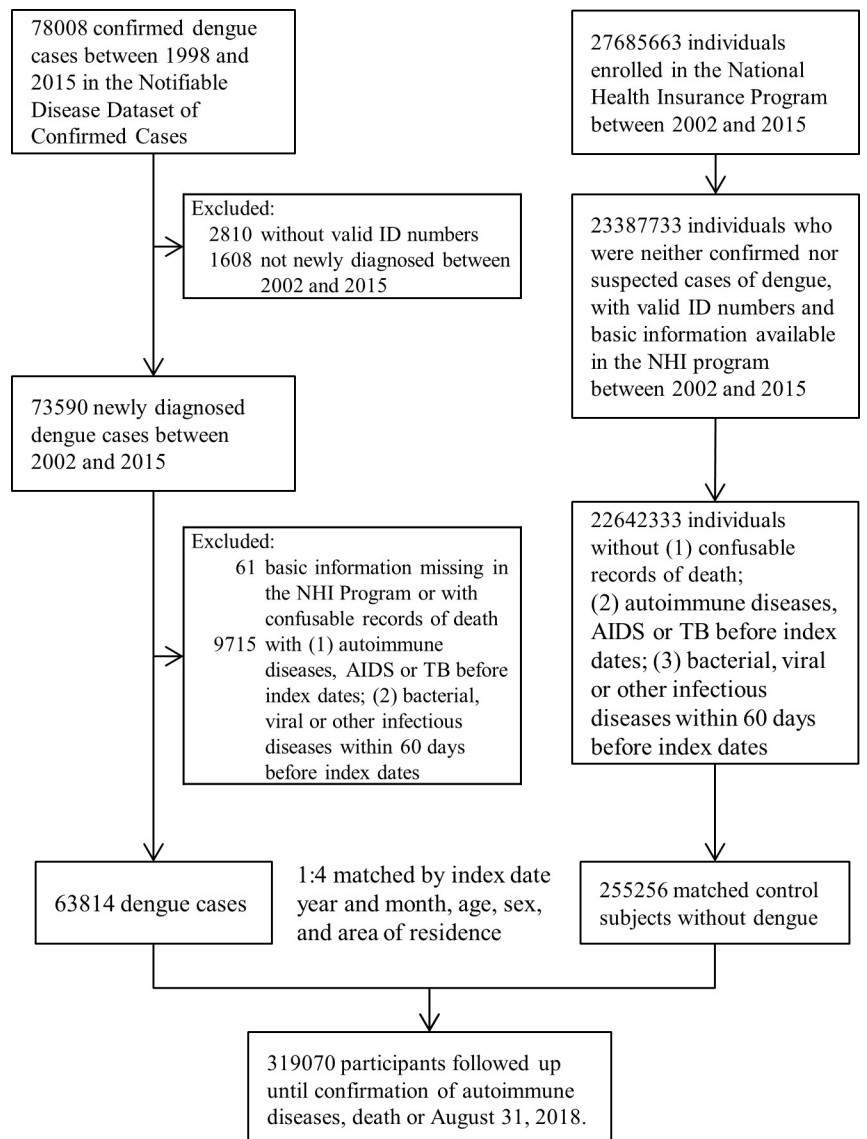

**Fig 1. Flow diagram depicting the selection of the study population.**

diseases than non-dengue controls after stratification by sex. Data on Type I DM, systemic sclerosis, inflammatory myopathy, Behcet's syndrome, systemic vasculitis, multiple sclerosis, autoimmune hemolytic anemia, inflammatory myopathy, pemphigus, and celiac disease cases were not shown because case numbers of less than three were not allowed to be exported under the regulations of the Health and Welfare Data Science Center of Taiwan to prevent re-identification.

This study evaluated the risk of overall and specific autoimmune diseases after DENV infection, stratified by sex. To account for multiple comparisons, the Bonferroni post-hoc method required a significance level of P<0.0012 (14 outcomes and analyses stratified by sex; total 42 tests in Table 3). After adjusting for age, sex, area of residence, urbanization level, monthly income level, and comorbidities using the sub-distribution hazard model, the risk of overall autoimmune diseases in the dengue group was significantly higher than that in the non-dengue group (aHR 1.16; P = 0.0002; Table 3). Stratified analyses by specific autoimmune diseases

**Table 1. Demographic and clinical characteristics of the dengue and non-dengue groups.**

| | Dengue cohort (n = 63814) | Non-dengue cohort (n = 255256) | Standardized mean difference |
|---|---|---|---|
| Sex | | | |
| Female | 31551 (49.4) | 126204 (49.4) | - |
| Male | 32263 (50.6) | 129052 (50.6) | - |
| Age (years) | 44.6 (20.0) | 44.6 (20.0) | - |
| 0–17 | 6806 (10.7) | 27224 (10.7) | - |
| 18–35 | 15528 (24.3) | 62112 (24.3) | - |
| 36–50 | 14111 (22.1) | 56444 (22.1) | - |
| 51–64 | 16368 (25.7) | 65472 (25.7) | - |
| ≥ 65 | 11001 (17.2) | 44004 (17.2) | - |
| Area of residence | | | |
| Tainan | 22407 (35.1) | 89628 (35.1) | - |
| Kaohsiung | 37770 (59.2) | 151080 (59.2) | - |
| Pingtung | 1587 (2.5) | 6348 (2.5) | - |
| Others | 2050 (3.2) | 8200 (3.2) | - |
| Income | | | |
| ≤ 1249 | 11869 (18.6) | 45365 (17.8) | 0.022 |
| 1250–21899 | 11631 (18.2) | 48622 (19.1) | 0.021 |
| 21900–34799 | 23930 (37.5) | 100214 (39.3) | 0.036 |
| ≥ 34800 | 16384 (25.7) | 61055 (23.9) | 0.041 |
| Urbanization | | | |
| 1 (most urbanized) | 23142 (36.3) | 60459 (23.7) | 0.285 |
| 2 | 25345 (39.7) | 82721 (32.4) | 0.155 |
| 3 | 12752 (20.0) | 68176 (26.7) | 0.154 |
| 4–7 (least urbanized) | 2575 (4.0) | 43900 (17.2) | 0.356 |
| Comorbidity | | | |
| Hypertension | 15709 (24.36) | 57080 (22.4) | 0.054 |
| DM | 7975 (12.5) | 28650 (11.2) | 0.040 |
| Dyslipidaemia | 13511 (21.2) | 45433 (17.8) | 0.087 |
| COPD | 8394 (13.2) | 29183 (11.4) | 0.053 |
| Cerebrovascular accident | 3819 (6.0) | 14401 (5.6) | 0.015 |
| Renal failure | 1797 (2.8) | 6147 (2.4) | 0.026 |
| Liver cirrhosis | 755 (1.2) | 2753 (1.1) | 0.010 |
| Ischaemic heart disease | 6565 (10.3) | 22401 (8.8) | 0.053 |
| Malignancy | 2645 (4.1) | 9599 (3.8) | 0.020 |

Data are represented as the number (%) or mean (standard deviation), unless otherwise stated. COPD = chronic obstructive pulmonary disease. DM = diabetes mellitus.

showed that DENV infection seemed to be associated with a higher risk of autoimmune thyroid disease, uveitis, and autoimmune encephalomyelitis at the significance level of 0.05; however, only autoimmune encephalomyelitis remained statistically significant after Bonferroni correction (aHR 2.72; $P < 0.0001$; Table 3). After stratification by sex, dengue infection was associated with a higher risk of autoimmune encephalomyelitis in males (aHR 32; $P = 0.0010$) and females (aHR 3.44; $P = 0.0001$; Table 3).

The Kaplan–Meier curves show that dengue patients had a significantly higher cumulative incidence of overall autoimmune diseases and autoimmune encephalomyelitis than the non-dengue cohort ($p<0.0001$, Fig 2A and 2B). Of 63,814 dengue patients, sixteen (0.025%) developed autoimmune encephalomyelitis within the first month after the index date, while none of

**Table 2. Incidence (per 10,000) of autoimmune diseases between the dengue and control groups.**

| Autoimmune disease | Dengue group | | | | | | Control group | | | | | |
|---|---|---|---|---|---|---|---|---|---|---|---|---|
| | Total (n = 63814) | | Male (n = 32263) | | Female (n = 31551) | | Total (n = 255256) | | Male (n = 129052) | | Female (n = 126204) | |
| | n | IR | n | IR | n | IR | n | IR | n | IR | n | IR |
| All | 809 | 27.73 | 349 | 23.89 | 460 | 31.59 | 2733 | 23.47 | 1161 | 19.93 | 1572 | 27.01 |
| Autoimmune thyroid disease | 236 | 8.03 | 64 | 4.35 | 172 | 11.71 | 782 | 6.67 | 183 | 3.12 | 599 | 10.22 |
| Uveitis | 179 | 6.08 | 88 | 5.98 | 91 | 6.19 | 554 | 4.72 | 267 | 4.56 | 287 | 4.89 |
| Psoriasis | 119 | 4.04 | 63 | 4.28 | 56 | 3.80 | 477 | 4.07 | 313 | 5.34 | 164 | 2.79 |
| Primary adrenocortical insufficiency | 69 | 2.34 | 32 | 2.17 | 37 | 2.51 | 260 | 2.21 | 127 | 2.17 | 133 | 2.26 |
| Ankylosing spondylitis | 66 | 2.24 | 50 | 3.40 | 16 | 1.09 | 271 | 2.31 | 162 | 2.76 | 109 | 1.85 |
| Autoimmune encephalomyelitis | 45 | 1.53 | 25 | 1.70 | 20 | 1.36 | 63 | 0.54 | 38 | 0.65 | 25 | 0.43 |
| Rheumatoid arthritis | 35 | 1.19 | 9 | 0.61 | 26 | 1.76 | 139 | 1.18 | 25 | 0.43 | 114 | 1.94 |
| Sjögren's syndrome | 21 | 0.71 | b | b | b | b | 99 | 0.84 | b | b | b | b |
| Systemic lupus erythematosus | 12 | 0.41 | 3 | 0.20 | 9 | 0.61 | 31 | 0.26 | 8 | 0.14 | 23 | 0.39 |
| Myasthenia gravis | 11 | 0.37 | 7 | 0.48 | 4 | 0.27 | 33 | 0.28 | 16 | 0.27 | 17 | 0.29 |
| Guillain–Barré syndrome | 10 | 0.34 | 4 | 0.27 | 6 | 0.41 | 30 | 0.26 | 17 | 0.29 | 13 | 0.22 |
| Inflammatory bowel diseases | 4 | 0.14 | b | b | b | b | 9 | 0.08 | b | b | b | b |
| Post-infectious arthritis | 4 | 0.14 | b | b | b | b | 17 | 0.14 | b | b | b | b |

Abbreviation: IR, incidence rate per 10,000 person-years.

[b] The cells left blank indicated that the number of events was less than three and were therefore not allowed to be exported under the regulations of the Health and Welfare Data Science Center of Taiwan to prevent re-identification.

[*] Data on Type I DM, systemic sclerosis, inflammatory myopathy, Behcet's syndrome, systemic vasculitis, multiple sclerosis, autoimmune hemolytic anemia, inflammatory myopathy, pemphigus, and celiac disease cases were not shown because case numbers were less than three.

**Table 3. Risk of autoimmune diseases after adjusting for competing mortality.**

| Autoimmune disease | Total | | Male | | Female | |
|---|---|---|---|---|---|---|
| | aHR[a] (95% CI)[b] | p value[c] | aHR[a] (95% CI)[b] | p value[c] | aHR[a] (95% CI)[b] | p value[c] |
| All | 1.16 (1.07–1.26) | 0.0002* | 1.18 (1.04–1.33) | 0.0094 | 1.15 (1.04–1.28) | 0.0092 |
| Autoimmune thyroid disease | 1.18 (1.02–1.37) | 0.0295 | 1.37 (1.02–1.83) | 0.0389 | 1.13 (0.95–1.34) | 0.1809 |
| Uveitis | 1.23 (1.03–1.46) | 0.0190 | 1.26 (0.98–1.61) | 0.0694 | 1.19 (0.94–1.51) | 0.1487 |
| Psoriasis | 0.97 (0.79–1.20) | 0.7856 | 0.77 (0.58–1.01) | 0.0594 | 1.38 (1.00–1.90) | 0.0472 |
| Primary adrenocortical insufficiency | 1.10 (0.84–1.44) | 0.5040 | 1.14 (0.76–1.70) | 0.5256 | 1.07 (0.74–1.55) | 0.7240 |
| Ankylosing spondylitis | 0.98 (0.74–1.28) | 0.8615 | 1.26 (0.91–1.73) | 0.1696 | 0.58 (0.34–0.97) | 0.0392 |
| Autoimmune encephalomyelitis | 2.72 (1.84–4.02) | <0.0001* | 2.32 (1.41–3.83) | 0.0010* | 3.44 (1.84–6.41) | 0.0001* |
| Rheumatoid arthritis | 0.99 (0.68–1.44) | 0.9456 | 1.62 (0.73–3.62) | 0.2386 | 0.87 (0.57–1.34) | 0.5339 |
| Sjögren's syndrome | 0.87 (0.53–1.41) | 0.5632 | 0.50 (0.06–4.09) | 0.5159 | 0.90 (0.54–1.48) | 0.6742 |
| Systemic lupus erythematosus | 1.86 (0.93–3.72) | 0.0817 | 1.87 (0.49–7.11) | 0.3606 | 1.86 (0.83–4.16) | 0.1334 |
| Myasthenia gravis | 1.36 (0.69–2.68) | 0.3744 | 1.80 (0.75–4.30) | 0.1874 | 0.96 (0.32–2.87) | 0.9445 |
| Guillain–Barré syndrome | 1.29 (0.61–2.72) | 0.5106 | 0.78 (0.26–2.35) | 0.6542 | 2.31 (0.81–6.58) | 0.1181 |
| Inflammatory bowel diseases | 2.01 (0.57–7.05) | 0.2748 | 2.31 (0.47–11.34) | 0.3024 | 1.11 (0.12–10.37) | 0.9256 |
| Post-infectious arthritis | 0.97 (0.33–2.82) | 0.9515 | 1.13 (0.32–4.01) | 0.8551 | 0.65 (0.08–5.04) | 0.6769 |

Abbreviations: aHR, adjusted hazard ratio; CI, confidence interval.

[a] Adjusted Cox proportional hazard model by sex, age, area of residence, income, urbanization, hypertension, diabetes mellitus, dyslipidemia, COPD, cerebrovascular accident, renal failure, liver cirrhosis, ischaemic heart disease, and malignancy.

[b] 95% CIs were not adjusted for multiple comparisons and thus cannot be directly used for hypothesis testing or inference.

[c] Post hoc adjustment for multiple comparisons by the Bonferroni method required a significance level of P<0.0012.

[*] Achieved statistical significance after Bonferroni correction for multiple comparisons (P<0.0012).

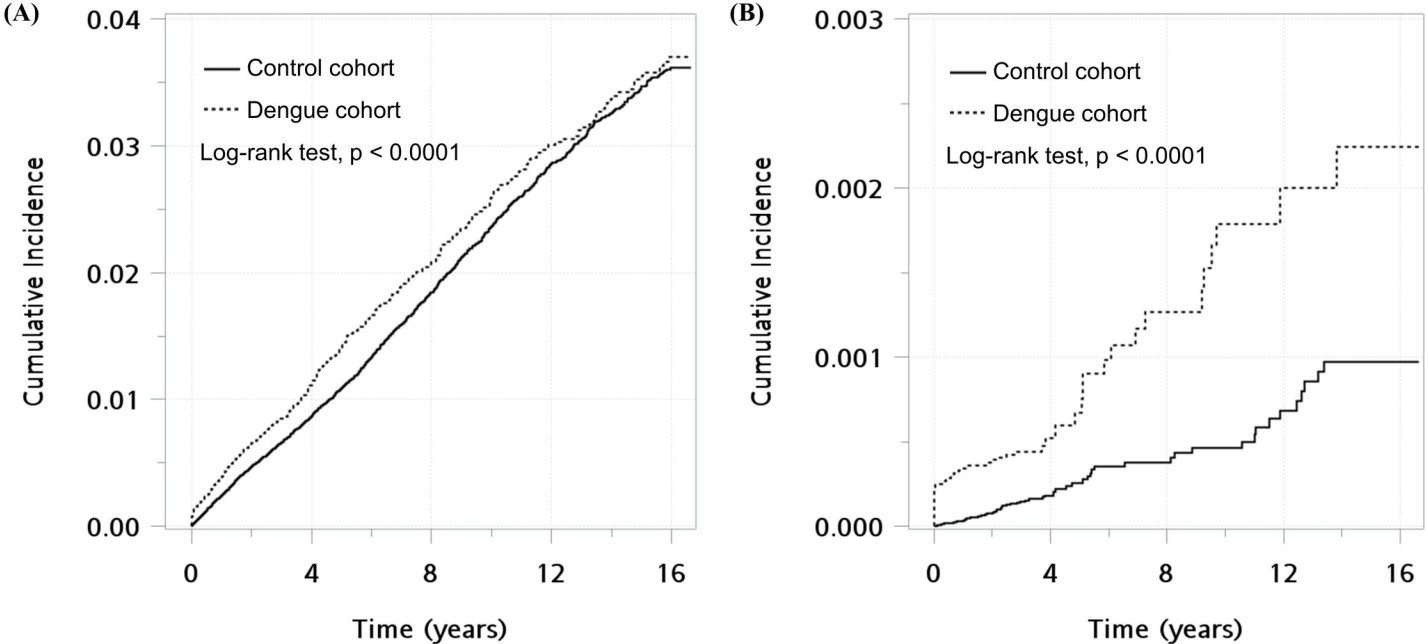

**Fig 2. Kaplan–Meier curves for the cumulative incidence of autoimmune disease (A) and autoimmune encephalomyelitis (B) between the dengue and non-dengue groups.**

the non-dengue controls had the disease during this period. Stratified analyses by follow-up time revealed that DENV infection was associated with a significantly higher risk of autoimmune encephalomyelitis in the first month after the onset of dengue symptoms (HR >9999, P < 0.0001) but not thereafter (aHR 1.76, P = 0.012) after considering multiple comparison correction.

## Discussion

This study suggests that the overall incidence of autoimmune diseases was slightly higher after DENV infection. Looking at specific autoimmune diseases, dengue was associated with an increased risk of autoimmune thyroid disease, uveitis, and autoimmune encephalomyelitis, but only autoimmune encephalomyelitis remained statistically significant after Bonferroni correction. Our results are very different from those reported by Li *et al.*, which suggested that dengue was associated with a wide spectrum of autoimmune diseases, including primary adrenocortical insufficiency, autoimmune encephalomyelitis, SLE, post-infectious arthritis, systemic vasculitis, myasthenia gravis, and multiple sclerosis [13]. In addition, another case-control study using NHIRD in Taiwan also showed that dengue was associated with SLE [19]. Three reasons may explain the difference. First, many dengue patients in the previous two studies were only suspected cases based on clinical symptoms, not laboratory-confirmed. The laboratory criteria for dengue diagnosis before 2015 in Taiwan included virus isolation, RNA detection by RT–PCR, a 4-fold increase in IgG titer in paired acute- and convalescent-phase samples, and dengue-specific IgM and IgG antibody detection in single serum samples. These tests were unavailable in most hospitals and clinics before 2014, and the samples needed to be delivered to certified laboratories approved by the Taiwan CDC. Therefore, the final laboratory-confirmed dengue diagnosis might have been delayed. The wide availability of non-structural protein 1 (NS1) rapid antigen tests from 2015 onwards shortened the confirmation time

and improved the accuracy of dengue diagnosis [20]. In addition, symptoms associated with autoimmune diseases, such as fever, skin rash, and thrombocytopenia, were very similar to the clinical presentation of dengue. In the Li *et al.* study, some patients with autoimmune diseases might have been initially misdiagnosed as having dengue during local dengue outbreaks and received the correct diagnosis later [13]. Second, multiple comparison problems were not considered in previous studies. Approximately 20 different outcomes were investigated, but the significance level was set at 0.05, which increased the probability of type I errors. Third, our study included inpatient and outpatient dengue cases, while the previous study only included hospitalized patients. However, we performed subgroup analyses of hospitalized dengue cases, revealing that dengue was only associated with autoimmune encephalitis but not other autoimmune diseases (S2 and S3 Tables).

Autoimmunity has been shown to be involved in dengue pathogenesis [21]. Autoantibodies produced during DENV infection can cross-react with several self-antigens on endothelial cells, platelets or coagulatory molecules, which may contribute to thrombocytopenia, coagulopathy, and vascular leakage in severe dengue [21–23]. Titers of DENV-induced autoantibodies reach peak levels in the acute phase, decline during the convalescent stage, and last for several months [21]. In addition, cases reports of SLE [8], systemic vasculitis [9], acute disseminated encephalomyelitis [10], neuromyelitis optica [11], and transverse myelitis [12] after dengue infection have been documented. However, our study found that DENV infection was only associated with short-term increased risk of autoimmune encephalomyelitis, but not other autoimmune diseases. The results suggest that some of the case reports might occur by chance and that the autoimmunity in dengue patients is transient.

Neuropathogenesis of DENV may occur in three ways: systematic or metabolic disturbance causing encephalopathy, direct central nervous system invasion (especially by DENV-2 and -3) causing mainly encephalitis, and autoimmunity-mediated complications including Guillain–Barré syndrome and acute disseminated encephalomyelitis (ADEM) [10,24]. Our study did not find a significant association between DENV and Guillain–Barré syndrome. Previous evidence suggests that ADEM, also known as postinfectious encephalomyelitis, results from a transient autoimmune response directed at myelin or other self-antigens, possibly by molecular mimicry or non-specific activation of auto-reactive T-cell clones [10,25]. A meta-analysis showed that 0.4% of dengue patients developed ADEM, which was much higher than in our study (0.025%) [10]. Our study was a population-based study including all laboratory-confirmed dengue cases diagnosed in hospitals and clinics, while previous studies only included patients treated in hospitals, which may explain the difference. The reported onset of neurological symptoms of ADEM ranges from day 3 to day 19 after initial dengue symptoms [10], which is consistent with our findings that the risk of autoimmune encephalomyelitis significantly increased within 30 days of DENV infection. Similar mosquito-borne single-stranded RNA flaviviruses, such as West Nile, Zika, and Chikungunya, have also been reported to be associated with autoimmune encephalomyelitis [26,27]. However, detailed mechanisms of dengue-associated autoimmune encephalomyelitis need further studies to clarify the pathogenesis.

Our study has several strengths. First, the sample size was large, with over 63,000 dengue patients and 255,000 controls, and the follow-up time was long. Second, this cohort study used national databases with a high coverage rate, minimizing selection bias caused by loss to follow-up. Third, all the dengue cases were laboratory-confirmed, minimizing the misclassification bias of the exposure status. Finally, our analyses made corrections for multiple testing by the Bonferroni method, which reduced the chance of false-positive findings. However, the Bonferroni correction is sometimes too conservative and may lead to a high rate of false negatives. Therefore, the Benjamini–Hochberg procedure to control the false discovery rate <0.05

was also performed [28], which yielded a critical P value of 0.0060. The number of hypotheses in Table 3 that reached the significance level remained the same, so our results were robust to Benjamini–Hochberg correction.

Several limitations in this study need to be addressed. First, although all dengue cases were laboratory-confirmed, some of the non-dengue controls might have been misclassified because some people with DENV infections were asymptomatic or only had mild symptoms and thus were not reported. However, since dengue has not been endemic in Taiwan and the overall incidence and seroprevalence remained low in most parts of Taiwan [29,30], this misclassification bias should be relatively low. Second, the data in NHIRD were originally collected for reimbursement purposes; therefore, information on some potential confounders that might contribute to autoimmune diseases, such as smoking habits, body weight, family history, chemical exposures, and genetics, was not available. Detailed clinical information such as duration of symptoms, severity, laboratory results and therapeutic interventions was also unavailable. Third, although the diagnosis of autoimmune diseases identified from the Registry for Catastrophic Illness should be 100% valid, the diagnosis of other autoimmune diseases might be less valid. To increase the accuracy of the diagnosis, we used hospital admission or at least three outpatient visits with relevant ICD-9-CM or ICD-10-CM codes to identify cases. We believed that the diagnosis of autoimmune encephalomyelitis should be relatively accurate because this diagnosis was usually made by neurologists, infectious disease specialists or rheumatologists in Taiwan; however, the validity of remaining autoimmune diseases was less certain. Fourth, data on the serotypes of dengue infection and whether patients had primary or secondary infection were unavailable; therefore, further investigation on the association between autoimmune diseases and DENV infection status could not be performed. Finally, several autoimmune diseases with fewer than three cases were unable to be reported because of the human subject protection regulations of the Health and Welfare Data Science Center, Ministry of Health and Welfare. Therefore, the risk of some autoimmune diseases could not be evaluated.

In conclusion, this large cohort study showed that dengue was associated with a short-term increased risk of autoimmune encephalomyelitis, a rare complication of dengue. The risk of other autoimmune diseases did not seem to increase after DENV infection.

## Supporting information

**S1 Table. List of ICD codes for identifying diseases in this study.**
(DOCX)

**S2 Table. Incidence (per 10,000) of autoimmune diseases between the hospitalized dengue cases and control groups.**
(DOCX)

**S3 Table. Risk of autoimmune diseases among hospitalized dengue cases after adjusting for competing mortality.**
(DOCX)

## Author Contributions

**Conceptualization:** Yu-Wen Chien.

**Data curation:** Pei-Fang Tsai, Yu-Ping Wang, Yu-Wen Chien.

**Formal analysis:** Hsin-I Shih, Pei-Fang Tsai, Yu-Ping Wang, Yu-Wen Chien.

**Funding acquisition:** Hsin-I Shih, Chia-Yu Chi, Yu-Wen Chien.

**Methodology:** Yu-Wen Chien.

**Project administration:** Chia-Yu Chi, Yu-Wen Chien.

**Resources:** Pei-Fang Tsai, Yu-Wen Chien.

**Software:** Yu-Ping Wang.

**Supervision:** Hsin-I Shih, Chia-Yu Chi, Yu-Wen Chien.

**Validation:** Hsin-I Shih, Yu-Ping Wang, Yu-Wen Chien.

**Writing – original draft:** Hsin-I Shih, Pei-Fang Tsai, Yu-Ping Wang, Yu-Wen Chien.

**Writing – review & editing:** Hsin-I Shih, Chia-Yu Chi, Yu-Wen Chien.

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
