## [Decision Letter · Decision Letter 0]

5 Oct 2022

Dear Dr. Chien,

Thank you very much for submitting your manuscript "Re-examination of the risk of autoimmune diseases after dengue virus infection: A population-based cohort study" for consideration at PLOS Neglected Tropical Diseases. As with all papers reviewed by the journal, your manuscript was reviewed by members of the editorial board and by several independent reviewers. In light of the reviews (below this email), we would like to invite the resubmission of a significantly-revised version that takes into account the reviewers' comments. 

We cannot make any decision about publication until we have seen the revised manuscript and your response to the reviewers' comments. Your revised manuscript is also likely to be sent to reviewers for further evaluation.

Sincerely,

William B Messer

Academic Editor

Elvina Viennet

Section Editor

Reviewer's Responses to Questions

**Key Review Criteria Required for Acceptance?**

**Methods**

-Are the objectives of the study clearly articulated with a clear testable hypothesis stated?

-Is the study design appropriate to address the stated objectives?

-Is the population clearly described and appropriate for the hypothesis being tested?

-Is the sample size sufficient to ensure adequate power to address the hypothesis being tested?

-Were correct statistical analysis used to support conclusions?

-Are there concerns about ethical or regulatory requirements being met?

Reviewer #1: This retrospective cohort study using laboratory-confirmed dengue cases evaluated the risk of autoimmune diseases after DENV infection. It was found that dengue was associated with an increased short-term risk of a rare disorder, “autoimmune” encephalomyelitis, but not with other autoimmune diseases. 

In the Author Summary the sentence “Autoantibodies produced during dengue infection can cross-react with several self-antigens…..” should change to: “Antibodies produced during dengue infection can cross-react with several self-antigens…..”

In the Introduction 2nd paragraph line 7-8 “autoimmune markers”, such as antinuclear antibody (ANA), immune complex (IC), and C-reactive protein (CRP)” IC and CRP are not autoimmune markers, ICs can be found in many chronic infectious diseases and CRP is an indicator of inflammation.

The term Reiter’s syndrome has to be changed to “post infectious arthritis”.

Reviewer #2: This is a well written study based on health records of dengue patients in Taiwan. The objectives are clearly articulated and the study design is appropriate. A strength of the study is the very large sample size. Correct statical analyses support the conclusion and indeed are essential to reach the conclusions. There are no ethical or regulatory concerns.

**Results**

-Does the analysis presented match the analysis plan?

-Are the results clearly and completely presented?

-Are the figures (Tables, Images) of sufficient quality for clarity?

Reviewer #1: s not clear to reviewer on what basis the authors use the term autoimmune encephalomyelitis for the syndrome following the DENV infection; did they tested the patients’ sera for specific autoantibodies? Also the length of encephalomyelitis symptoms , the severity of the disorder and the therapeutic interventions to combat the encephalomyelitis should be presented .

The term Reiter’s syndrome should be changed to “post infectious arthritis”

Reviewer #2: The results are clearly presented. Dengue patients had a slightly higher risk of overall autoimmune diseases than non-dengue controls (aHR 1.16; P =0.0002). Only autoimmune encephalomyelitis remained statistically significant after Bonferroni correction for multiple testing (aHR 2.72; P < 0.0001). All the Table are presently clearly.

**Conclusions**

-Are the conclusions supported by the data presented?

-Are the limitations of analysis clearly described?

-Do the authors discuss how these data can be helpful to advance our understanding of the topic under study?

-Is public health relevance addressed?

Reviewer #1: AS ABOVE

Reviewer #2: The conclusion of the study are justified and limitations are described appropriately. The discussion is appropriate and the public health relevance is addressed appropriately.

**Editorial and Data Presentation Modifications?**

Reviewer #1: As Above

Reviewer #2: The results are different than a previous smaller study that identified several autoimmune disease that were increased in hospitalized dengue patients. The authors do a good job of describing the differences in the studies, bit I think imless they did the sub-analysis of only hospitalized patients they should explicitly state that hospitalization might mean more severe dengue and that more severe dengue might in fact lead to autoimmune disease.

**Summary and General Comments**

Reviewer #1: As Above

Reviewer #2: This is a solid study. The authors should clarify that it remains possible that more severe dengue can lead to autoimmune disease.

PLOS authors have the option to publish the peer review history of their article (what does this mean?). If published, this will include your full peer review and any attached files.

Reviewer #1: Yes: Haralampos M. Moutsopoulos

Reviewer #2: No
---

## [Decision Letter · Decision Letter 1]

30 Jan 2023

Dear Dr. Chien,

We are pleased to inform you that your manuscript 'Re-examination of the risk of autoimmune diseases after dengue virus infection: A population-based cohort study' has been provisionally accepted for publication in PLOS Neglected Tropical Diseases.

Best regards,

William B Messer

Academic Editor

Elvina Viennet

Section Editor

Reviewer's Responses to Questions

**Key Review Criteria Required for Acceptance?**

**Methods**

-Are the objectives of the study clearly articulated with a clear testable hypothesis stated?

-Is the study design appropriate to address the stated objectives?

-Is the population clearly described and appropriate for the hypothesis being tested?

-Is the sample size sufficient to ensure adequate power to address the hypothesis being tested?

-Were correct statistical analysis used to support conclusions?

-Are there concerns about ethical or regulatory requirements being met?

Reviewer #1: The authors responded satisfactorily to reviewers' suggestions

Reviewer #2: This revised manuscript meets all review criteria for accepotance.

**Results**

-Does the analysis presented match the analysis plan?

-Are the results clearly and completely presented?

-Are the figures (Tables, Images) of sufficient quality for clarity?

Reviewer #1: The analysis presented match the analysis plan

The results are clearly and completely presented

The figures (Tables, Images) are of sufficient quality for clarity

Reviewer #2: The results are clearly presented

**Conclusions**

-Are the conclusions supported by the data presented?

-Are the limitations of analysis clearly described?

-Do the authors discuss how these data can be helpful to advance our understanding of the topic under study?

-Is public health relevance addressed?

Reviewer #1: (No Response)

Reviewer #2: The conclusions are supported.

**Editorial and Data Presentation Modifications?**

Reviewer #1: (No Response)

Reviewer #2: none

**Summary and General Comments**

Reviewer #1: (No Response)

Reviewer #2: I have np further comments.

PLOS authors have the option to publish the peer review history of their article (what does this mean?). If published, this will include your full peer review and any attached files.

Reviewer #1: No

Reviewer #2: No

---

## [Editor Report · Acceptance letter]

21 Feb 2023

Dear Dr. Chien,

We are delighted to inform you that your manuscript, "Re-examination of the risk of autoimmune diseases after dengue virus infection: A population-based cohort study," has been formally accepted for publication in PLOS Neglected Tropical Diseases.

Best regards,

Shaden Kamhawi

co-Editor-in-Chief

Paul Brindley

co-Editor-in-Chief
